# Anti-Fibrotic Potential of Angiotensin (1-7) in Hemodynamically Overloaded Rat Heart

**DOI:** 10.3390/ijms24043490

**Published:** 2023-02-09

**Authors:** Matus Sykora, Vojtech Kratky, Libor Kopkan, Narcisa Tribulova, Barbara Szeiffova Bacova

**Affiliations:** 1Centre of Experimental Medicine, Slovak Academy of Sciences, Institute for Heart Research, 841 04 Bratislava, Slovakia; 2Center for Experimental Medicine, Institute for Clinical and Experimental Medicine, 140 21 Prague, Czech Republic; 3Department of Nephrology, First Faculty of Medicine, Charles University and General University Hospital in Prague, 128 08 Prague, Czech Republic

**Keywords:** heart failure, aortocaval fistula, extracellular matrix, connexin 43, angiotensin (1-7)

## Abstract

The extracellular matrix (ECM) is a highly dynamic structure controlling the proper functioning of heart muscle. ECM remodeling with enhanced collagen deposition due to hemodynamic overload impairs cardiomyocyte adhesion and electrical coupling that contributes to cardiac mechanical dysfunction and arrhythmias. We aimed to explore ECM and connexin-43 (Cx43) signaling pathways in hemodynamically overloaded rat heart as well as the possible implication of angiotensin (1-7) (Ang (1-7)) to prevent/attenuate adverse myocardial remodeling. Male 8-week-old, normotensive Hannover Spraque–Dawley rats (HSD), hypertensive (mRen-2)27 transgenic rats (TGR) and Ang (1-7) transgenic rats (TGR(A1-7)3292) underwent aortocaval fistula (ACF) to produce volume overload. Five weeks later, biometric and heart tissue analyses were performed. Cardiac hypertrophy in response to volume overload was significantly less pronounced in TGR(A1-7)3292 compared to HSD rats. Moreover, a marker of fibrosis hydroxyproline was increased in both ventricles of volume-overloaded TGR while it was reduced in the Ang (1-7) right heart ventricle. The protein level and activity of MMP-2 were reduced in both ventricles of volume-overloaded TGR/TGR(A1-7)3292 compared to HSD. SMAD2/3 protein levels were decreased in the right ventricle of TGR(A1-7)3292 compared to HSD/TGR in response to volume overload. In parallel, Cx43 and pCx43 implicated in electrical coupling were increased in TGR(A1-7)3292 versus HSD/TGR. It can be concluded that Ang (1-7) exhibits cardio-protective and anti-fibrotic potential in conditions of cardiac volume overload.

## 1. Introduction

Heart failure (HF) is a clinical syndrome characterized by cardiac dysfunction due to structural abnormalities of the myocardium that results in the inability of the heart to eject a sufficient amount of the blood to the circulation, to cover the metabolic needs of the body. HF is a major cause of morbidity and mortality and represents an extensive health and economic burden with a huge impact due to high treatment costs, frequent hospitalizations and poor quality of life [1]. Pressure and volume overload of the heart is critically involved in the pathogenesis of HF. Left ventricular pressure overload is a severe stressor induced by systemic hypertension or aortic stenosis, promoting HF. At the same time, valvular regurgitation lesions (such as aortic and mitral insufficiency) cause volume overload resulting in myocardial remodeling and HF. Both pressure and volume loading contribute to the adverse remodeling of the extracellular matrix (ECM), impacting both the mechanical and electrical properties of cardiomyocytes [2,3,4].

The ECM as a mechanical support system maintaining tissue integrity also serves as a signaling hub. This dynamic structure transmits signal cascades critical for proper cell function, it is a reservoir of growth factors and proteases that modulate various repair processes and can be activated in the case of pathophysiology [5,6,7,8].

Electrical and metabolic communication is sustained directly via connexin-based (Cx) gap junction channels (GJC), essential for ensuring the transmission of electrical impulses and coordinated contractile and pumping activity. The prevalent isoform of Cx expressed in the heart is Cx43. Impairment of Cx43 GJC promotes the development of electrical conduction disorders, which are the basis of heart diseases, malignant arrhythmias and HF [9,10,11,12]. Therefore, the understanding and subsequent modulation of molecular pathways implicated in mechanical and electrical heart remodeling can lead to the development of new pharmacological and non-pharmacological strategies for cardioprotection.

The renin–angiotensin–aldosterone system (RAAS) is a well-known crucial factor involved in cardiac remodeling in both pressure and volume overload conditions. On the other hand, experimental studies point out the benefit of the non-canonical pathway of angiotensin-converting enzyme 2 (ACE2)-Ang (1-7)-Mas receptor, which counteracts the adverse effects of the RAAS pathway. The cardioprotective effects of angiotensin (1-7) appear to be mediated by different signaling pathways [13]. Studies have shown that the administration of Ang (1-7), during hemodynamic overload of the heart reduces myocyte hypertrophy and cardiac fibrosis [14,15]. Deletion for ACE2, on the other hand, has been shown to cause an increase in perivascular and interstitial fibrosis, ventricular dilatation, decreased intrinsic myocardial contractility and increased cardiac remodeling, as well as a general deterioration of ventricular functions up to increased mortality [16,17,18]. Thus, we aimed to explore possible implication of Ang (1-7) on myocardial ECM and Cx43 in conditions of cardiac volume overload.

## 2. Results

### 2.1. Biometric Parameters of Rats Affected by ACF and Ang (1-7)

Body weight did not differ between individual rat strains. Heart weight (HW) index per tibia length also did not differ between the HSD ACF and TGR ACF strains. However, HW was significantly reduced in TGR(A1-7)3292 ACF compared to HSD ACF or TGR ACF rats. Left ventricular weight index was increased in TGR ACF and decreased in TGR(A1-7)3292 ACF vs. HSD ACF. There was an increase in left ventricular weight in TGR(A1-7)3292 ACF compared to TGR(A1-7)3292. We registered a reduction right ventricular weight index in TGR(A1-7)3292 ACF compared to HSD ACF or TGR ACF while there was an increase in TGR(A1-7)3292 ACF compared to TGR(A1-7)3292 (Figure 1).

### 2.2. Myocardial Markers of Oxidative Stress and Fibrotic Activity

Oxidative stress plays an important role in heart failure, including ACF. The increased synthesis and secretion of ROS leads to a disturbance of the oxidative balance, which in turn stimulates many signaling pathways that regulate various processes, including the promotion of cell proliferation and migration and the secretion of ECM [19]. The levels of TBARS, as a marker of oxidative stress, decreased in the left ventricle in the TGR(A1-7)3292 group compared to the HSD ACF group and in the right ventricle, and also in the TGR(A1-7)3292 group, but compared to the TGR ACF group (Figure 2).

Representative microscopic images of hematoxylin–eosin-stained myocardial tissue (apex) demonstrates the prevalent population of enlarged cardiomyocytes and fibrosis in TGR ACF versus HSD ACF rats (Figure 3). Hematoxylin–eosin staining (Figure 3A) revealed in TGR ACF rats an increased focal area infiltrated with polymorphonuclears (arrows), the histopathological feature of the hypertrophy or fibrosis. Moreover, hydroxyproline evaluation of collagen deposition (Figure 3B,C) revealed increased levels of collagen content in TGR ACF rats. Ang (1-7) notably suppressed polymorphonuclears and hydroxyproline content in TGR ACF rats indicating its antifibrotic potential. Hydroxyproline is a breakdown product of collagen occurring mainly in tissue fibrosis and overall cardiac remodeling expected in ACF-induced heart failure [20]. Hydroxyproline was increased in both heart ventricles in TGR ACF vs. HSD ACF. However, there was a significant decrease in hydroxyproline content in the right ventricle of TGR(A1-7)3292 ACF vs. TGR ACF (Figure 3). The sum of the results of HE staining and biochemical analysis of hydroxyproline points to the fact that Ang (1-7) in our HF model significantly reduces the level of fibrosis, especially in the right ventricle of the heart compared to TGR and normalizes them to the level of HSD control.

### 2.3. Determination of Myocardial MMP-2 Activity and Protein Levels

MMP-2 is a metalloproteinase that has an important role in the process of ECM remodeling, and its activation could be related to structural changes in cardiac ECM. There was a significant decrease in MMP-2 enzymatic activity in both heart ventricles of TGR ACF and TGR(A1-7)3292 ACF rats (Figure 4) compared to basal MMP-2 activity in normotensive HSD ACF rats (Figure 4). There were no significant changes of MMP-2 protein abundance in the left ventricle of experimental rats. (Figure 4). However, MMP-2 protein levels were reduced in the right ventricle of TGR ACF and TGR(A1-7)3292 ACF as well as TGR(A1-7)3292 when compared to HSD ACF.

The mechanism of MMP-2 could differ between ventricles since they have a different function. These changes could also be more pronounced in our case because the right ventricle is much more affected by hemodynamic overload [21]. Unfortunately, the tendency of protein is slightly similar or shows no change (HSD ACF vs. TGR ACF); thus, it is difficult to attribute it to a different mechanism. We assume that this discrepancy between the ventricle could be attributed to the higher hemodynamic effect on right ventricle.

### 2.4. Determination of SMAD Protein Levels Implicated in Fibrosis

Increased expression of SMAD2 and SMAD3 was demonstrated mainly in fibroblasts infiltrating fibrotic and remodeling hearts [22]. We observed a decrease in the protein level of the profibrotic sum of SMAD2/3 in the left ventricle of TGR ACF versus HSD ACF and an increase in TGR(A1-7)3292 versus TGR ACF. In contrast, the right ventricle exhibited a reduction in the sum of SMAD2/3 in the TGR(A1-7)3292 ACF group compared to the HSD ACF and TGR ACF groups (Figure 5).

### 2.5. Determination of PKC Protein Levels

Protein levels of PKCα, implicated in the myocardial hypertrophy [23], were decreased in the left and right heart ventricles in TGR(A1-7)3292 ACF strain compared to HSD ACF or TGR ACF (Figure 6). Protein levels of pro-apoptotic, pro-hypertrophic and pro-fibrotic PKCδ [23] are demonstrated in Figure 6. There was a decline in PKCδ protein abundance in the TGR ACF and TGR(A1-7)3292 groups compared to HSD ACF as well as in TGR(A1-7)3292 ACF rats vs. TGR ACF. One of the protein kinases which directly phosphorylates Cx43 at Serine 368 is PKC-ε [24]. Contrary to PKCα and PKCδ, PKCε protein levels were increased in the TGR ACF and TGR(A1-7)3292 ACF rats vs. HSD ACF as well as in TGR(A1-7)3292 ACF rats vs. TGR ACF. The same trend of protein levels was observed in both left and right heart ventricles (Figure 6).

### 2.6. Myocardial Cx43 Protein Levels and Topology

Structural remodeling is associated with changes of Cx43 (protein levels and localization) responsible for electrical instability leading to increased risk of life-threatening arrhythmias. The protein levels of Cx43 in experimental rat hearts are demonstrated in (Figure 7). There was a significant increase in Cx43 protein abundance in the TGR(A1-7)3292 ACF group vs. HSD ACF/TGR ACF in both heart ventricles. However, ACF in the TGR(A1-7)3292 reduced protein abundance of Cx43 in the left as well as right ventricle. Protein levels of functional phosphorylated form of Cx43 were significantly increased in both heart ventricles of TGR hypertensive strain compared to normotensive HSD ACF rats. (Figure 7). However, ACF in the TGR(A1-7)3292 reduced the protein abundance of the phosphorylated form of Cx43 in the left as well as right ventricle (Figure 7). Representative microscopic images of Cx43 cardiomyocyte distribution in the myocardium (apex) of experimental rats are shown in Figure 7. There is prevalent localization at the intercalated discs of the cardiomyocytes in normotensive HSD ACF. Notably, hypertensive TGR strains exhibit pronounced Cx43 distribution on lateral sides of the cardiomyocytes. Quantitative image analysis revealed a significant increase in the immunofluorescence signal in TGR(A1-7)3292 rats as well as with ACF. Total integral optical density per area (IOD) of Cx43 was similar to Cx43 protein levels significantly elevated in the myocardium (apex) of transgenic rats with increased expression of TGR(A1-7)3292 as well as in TGR(A1-7)3292 ACF.

## 3. Discussion

In order to detect the possible structural remodeling (hypertrophy) of the heart chambers after ACF induction and to monitor the influence of Ang (1-7), the biometric parameters were registered. The rat body weight was the same in all strains of rats with ACF. According to the literature, an increase in the whole heart, left ventricle and right ventricle weight normalized to the tibia length is attributed to cardiac hypertrophy [25,26,27]. We did not observe a difference in the heart weight between the HSD and TGR strains after ACF, the TGR(A1-7)3292 strain and after ACF as well. This may indicate a certain antihypertrophic effect of Ang (1-7), which was also observed in other studies [28].

Right ventricular weight imitated the results of the whole heart. Since the ACF model acutely affects mostly the right side of the heart, and based on the literature, it is logical that the right ventricular weight increases and eccentrically hypertrophies [29]; we also recorded a positive effect of Ang (1-7) in the right ventricle of TGR(A1-7)3292 strain vs. HSD and TGR after ACF. The biometric index of the left ventricle exhibited the trend of right ventricle probably due to the adaptation of the heart to volume overload in the compensatory phase of HF. On the other hand, we noted an increase in the weight of the left ventricle of TGR compared to HSD rats, which is due to the fact that the TGR strain has moderately increased blood pressure and the remodeling of the left ventricle takes place due to pressure overload even before the induction of ACF, as well as after the induction of ACF. The TGR(A1-7)3292 strain had a significantly smaller weight of the left ventricle after ACF than HSD or TGR strains. Altogether, biometric parameters indicate that Ang (1-7) influenced the size of the heart most likely by affecting cardiomyocyte growth.

We focused on the analysis of oxidative stress markers, such as TBARS, which reflects lipid peroxidation. We did not detect any alterations in heart tissue TBARS levels between individual groups or a positive or negative effect of Ang (1-7), similar to other studies [30,31]. While a positive antioxidant effect of Ang (1-7) in the kidney has been observed [32]. Furthermore, the positive influence of Ang (1-7) on the oxidation state in epididymal fat during a high-fructose diet was observed [33].

Another analyzed marker pointing out structural remodeling is hydroxyproline, a collagen cleavage product occurring mainly in tissue fibrosis [20]. We noticed an increasement of hydroxyproline content in the TGR group versus HSD after ACF. It is consistent with the fact that volume overload (already in period from 4 to 15 weeks after ACF induction) is characterized by an increased accumulation of the extracellular matrix proteins leading to cardiac fibrosis [25,34]. We observed that Ang (1-7) had no effect on the amount of hydroxyproline in the TGR(A1-7)3292 strain after ACF compared to HSD with ACF in neither the left or the right ventricle. The result in the TGR(A1-7)3292 rats’ strain is interesting, because even ACF did not cause an increase in hydroxyproline, pointing out a cardioprotective effect [25,35].

Hydroxyproline is directly linked to MMP-2 activity and protein levels. MMP-2 cause the splitting of components of the extracellular matrix, especially collagen, and thereby affect ECM remodeling and fibrosis. Both activity and protein abundance of MMP-2 were increased by ACF in the compensated or decompensated phase of HF, which is a response to the increasing need to synthesize collagen [36]. It can be considered as a compensatory mechanism in adaptation to the volume overload, which disrupts the myocardial homeostasis [25,27,37,38]. Consistent with other studies, we detected higher protein levels and activity of MMP-2 in the RV of normotensive HSD rats, and lower protein levels and activity in both hypertensive TGR and TGR(A1-7)3292 rats after ACF. Ang (1-7) had no effect on MMP-2. We detected lower MMP-2 activity in the left ventricle of TGR versus HSD after ACF, but the protein abundance was not changed. Noteworthy, ACF did not affect MMP-2 in the TGR(A1-7)3292 strain, which implies the possible anti-fibrotic effect of Ang (1-7). In our opinion, the decrease in MMP-2 activity or its protein levels may be caused by the reduction of mean arterial pressure due to ACF in hypertensive TGR rats [27] and adaptation of the myocardium to hypertension. However, our hypothesis is associated with the ACF model, while data from hypertensive individuals suffering from volume overload are missing.

The implication of intracellular SMAD2/3 signaling involved in various hypertrophic and fibrotic pathways was also observed in HF models [39]. However, we did not detect alteration of SMAD2/3 between the HSD and TGR groups after ACF in the left or right heart ventricles. It may indicate that volume overload due to ACF is a stronger factor than hypertension in the compensatory phase. Ang (1-7) normalized SMAD2/3 protein expression in the right ventricle, compared to the HSD ACF and TGR ACF groups. This effect can be similar to the effect of simvastatin reported by Tang et al. (2021) [39] and considered as a certain form of cardioprotection.

Protein kinases C is a family of kinases involved in intracellular signaling during cardiovascular remodeling due to pressure or volume overload. In the heart, PKCε and PKCδ are the most expressed PKC isoenzymes, which are contradictory in cardioprotection, with the positive role of PKCε and the inhibitory role of PKCδ. In the process of hypertrophy, PKCε and PKCδ move in the same direction. PKCε induces ventricular hypertrophy via cardiomyocyte growth through phosphorylation of target proteins. PKCδ is mainly associated with ventricular hypertrophy under pathologies. Decline in PKCε and an increment in PKCδ protein levels, which has been found in several models of HF [40,41]. PKCα is responsible for cardiomyocyte growth, stimulates myocardial hypertrophy and reduces the apoptotic burden. Overexpression of PKCα protein causes a decrease in the contractile force, whereas inhibition of PKCα results in an increase in cardiac contractile performance [42,43]. Consistent with previous studies, PKCδ protein levels were decreased due to ACF reflecting possible adaptation of the myocardium to hypertension in TGR versus HSD [27]. In contrast to PKCδ, PKCε was increased in HSD and TGR groups after ACF. They act antagonistically, as also shown by others [44]. In the left ventricle, the expression of PKCε was significantly increased by the influence of Ang (1-7) in contrast to the right ventricle. It may be due to the fact that the ACF model affects the right ventricle more significantly. We detected a similar change in PKCα in the left and right ventricles, except for a significant decrease in the right ventricle in TGR compared to HSD. We did not observe any change in the left ventricle between normotensive and hypertensive animals. The TGR(A1-7)3292 strain had significantly reduced PKCα after ACF that can be attributed to the fact that Ang (1-7) has an effect through intracellular signaling on the development of hypertrophy and the contractile force. Likewise in other studies [42,43], we also did not detect in this rat strain an increase in PKCα after ACF.

Furthermore, we examined Cx43 as a protein that forms communicating channels between adjacent cardiomyocytes that are responsible for the transmission of electrical impulses. Abnormalities of Cx43 are mostly associated with electrical instability of the myocardium and thereby with an increased risk of malignant arrhythmias. Moreover, reduced expression of Cx43 promotes ECM remodeling and fibrosis [45,46]. Previously, Cx43 changes were analyzed in the decompensatory phase, not in the compensatory phase of HF [47]. Guggilam et al. (2013) [47] observed a significant decrease in Cx43 protein abundance, a prolongation of the QTc interval, reduced propagation of Ca^2+^ waves, which ultimately promote the occurrence of arrhythmias, and reduced cardiac contractility and systolic function of the heart. We did not detect alteration in Cx43 protein levels in the left ventricle after ACF in normotensive and hypertensive (HSD/TGR) rats, in contrast to the right ventricle, where we observed an increased protein levels of Cx43 in TGR after ACF versus HSD-ACF. We can explain the difference between HSD and TGR rats after ACF by the adaptation and partial normalization of pressure after ACF in hypertensive rats. Consistent with the study of Cao et al. (2018) [48], we also detected a significant increase in Cx43 in the TGR(A1-7)3292 strain after ACF compared to HSD and TGR after ACF. We demonstrated this cardioprotective effect of Ang (1-7) in both the left and right ventricles. A similar trend was observed in the phosphorylated form of Cx43, which is considered to be the active form of Cx43 [49,50]. PKCε, as one of the protein kinases responsible for the phosphorylation of Cx43 at serine 368, is associated with antiarrhythmic rat phenotype [51,52]. We found that pCx43 copies the protein levels of PKCε in the HSD and TGR groups after ACF in both the left and right ventricles. Even in the group of TGR(A1-7)3292 strain, we could see a decrease in pCx43 protein levels after ACF. Similar with total Cx43, we detected the cardioprotective effect of Ang (1-7) compared to HSD and TGR after ACF. Several studies have revealed a regulation of Cx43 phosphorylation at S368 by multiple PKC isozymes, not only by PKCε. For instance, PKCδ, a novel PKC isoform, has been shown to phosphorylate S368 leading to gap junction internalization and degradation through the proteasomal and lysosomal pathway [53,54]. Since in TGR(A1-7)3292 ACF rats (vs. TGR ACF) PKCδ was decreased, we can assume that an increase in Cx43 at serine 368 can be a result of decreased Cx43 internalization and degradation.

If we look at the results comprehensively, it can be concluded that Ang (1-7) exhibits cardio-protective and anti-fibrotic potential in conditions of cardiac volume overload. A limitation of this work is the absence of sham control groups in the HSD and TGR rats. In the future, it would be certainly interesting to analyze these cardioprotective effects after a longer time, or direct administration of Ang (1-7), after ACF in both normotensive and hypertensive animals.

## 4. Materials and Methods

### 4.1. Experimental Design

TGR(A1-7)3292 rats show specific testicular expression of a cytomegalovirus promoter-driven transgene that results in a doubling of circulating Ang (1-7) compared to non-transgenic control rats [55]. The hypertensive (mRen-2)27 transgenic rat strain (TGR) shows strong expression of the murine Ren-2 transgene in extrarenal tissues, which induces and maintains hypertension through conventional angiotensin II (Ang II), and hypertension is readily controlled by inhibition of the renin–angiotensin system (RAS). This transgenic rat strain is now widely used to study a variety of conditions related to tissue RAS activation, including angiogenesis, cytokine activation, profibrotic and inflammatory pathologies, thus contributing to the understanding of the underlying processes causing severe hypertension [56,57]. In the experimental model, 20 male TGR(A1-7)3292 laboratory rats, 10 male Hannover Sprague–Dawley (HSD) laboratory normotensive rats and 10 male (mRen-2)27 transgenic laboratory hypertensive rats (TGR) at the age of 8 weeks were used. HSD rats are standardly used as a normotensive control for TGR rats. The experimental animals came from the accredited laboratory breeding of the Center for Experimental Medicine, Institute of Clinical and Experimental Medicine in Prague, Czech Republic. All animal experiments were approved on 26 June 2017 by the Animal Care and Use Committee of the Institute for Clinical and Experimental Medicine, Prague; project number 50/2017; in accordance with guidelines and practices established by the Directive 2010/63/EU of the European Parliament on the Protection of Animals Used for Scientific Purposes. Laboratory animals TGR(A1-7)3292 were divided into two groups with sham surgery and ACF surgery, HSD and TGR with ACF surgery (Table 1). The animals were fed a standard laboratory diet, which, was available to them ad libitum, as well as drinking water. They were kept in air-conditioned rooms with a constant temperature of 22–24 °C, humidity of 40–60% and with a regular light regime of 12 h of darkness and 12 h of light. The ACF operation was performed under general anesthesia induced by isoflurane. After exposing the abdominal aorta and inferior vena cava, the aorta was occluded for 30 s in the area between the renal arteries and the iliac bifurcation. ACF was created by inserting a needle (diameter 1.2 mm) through the abdominal aorta into the inferior vena cava. The injection site was then sealed with cyanoacrylate tissue adhesive [58]. Five weeks after ACF induction, rats in the “compensated HF” phase were euthanatized by decapitation.

### 4.2. Western Blot Assay

As was described previously in our publications [51,59], frozen left ventricular heart tissue was powdered, extracted in SDS lysis buffer (20%, 10 mmol/L EDTA, 100 mmol/L Tris, pH 6.8) and diluted in Laemmli buffer. Proteins were separated in 10% SDS-polyacrylamide gel and transferred to a nitrocellulose membrane (0.2 m pore size, Advantec, Tokyo, Japan). Membranes were blocked in 5% low-fat milk and then incubated with primary antibodies (Table 2) and horseradish peroxidase-linked secondary antibody (Table 2). Enhanced chemiluminescence was used for detection of proteins, which were consequently in triplicate quantitated using Carestream Molecular Imaging Software (version 5.0, Carestream Health, New Haven, CT, USA). For protein normalization GAPDH protein was used.

### 4.3. Gelatine Zymography for MMP-2 Activity Assessment

Samples were prepared and separated in 10% gels copolymerized with gelatin (2 mg/mL) same as in the Western blot method, but in non-reducing conditions. Gels were soaked in washing buffer (50 mmol/L Tris-HCl, 2.5% Triton X-100, pH 7.4) and incubated in developing buffer at 37 °C (50 mmol/L Tris-HCl, 10 mmol/L CaCl_2_, 1.25% Triton X-100, pH 7.4). After overnight incubation, gels were stained in stain solution (1% Coomassie Brilliant Blue G-250 dissolved in a solution containing 10% acetic acid and 40% methanol) and distained with a solution containing 10% acetic acid and 40% methanol. Finally, transparent bands on a dark blue background appeared. Bands considered as enzymatic activities of MMP-2 were densitometric quantified by Carestream Molecular Imaging Software (version 5.0, Carestream Health, New Haven, CT, USA) [60].

### 4.4. Cx43 Immunostaining and Quantitative Analysis

For Cx43 immunodetection, 10 µm thick cryosections of myocardial apex tissue were used. According to our previous publications [59,61], cryosections were fixed in ice-cold methanol, permeabilized in 0.3% Triton X-100, blocked in solution of 1% bovine serum albumin and incubated with primary anti-Cx43 antibody (diluted 1:500, CHEMICON International, Inc., Temecula, CA, USA, #MAB 3068 and secondary antibody with FITC-fluorescein isothiocyanate (diluted 1:500, Jackson Immuno Research Labs, West Grove, PA, USA, #111-095-003). For actin filaments, visualization cryosections were stained with phalloidin (Sigma-Aldrich, St. Louis, MO, USA, #P 2141). Microscopic images were captured by Zeiss Apotome 2 microscope (Carl Zeiss, Jena, Germany). Quantification of Cx43 immunofluorescence signal was performed on a 15 randomly selected myocardial area (per heart) and expressed as total integral optical density per area (IOD) [62].

### 4.5. Collagen Content Determination by Hydroxyproline Assay

The fibrosis marker, hydroxyproline, was spectrophotometric evaluated as was described previously [59]. Myocardial tissue was hydrolyzed in 6 M HCl very briefly, dried and incubated in the solution of chloramine T-acetate-citrate buffer (pH 6.0). Finally, to stop oxidation reaction, Ehrlich’s reagent solution was added. Hydroxyproline was measured spectrophotometrically at 550 nm and expressed in mg per total weight of the LV and RV.

### 4.6. Measurement of Malondialdehyde Level

According to Shlafer and Shepard (1984), with some modifications [63,64], approximately 40 µL of tissue homogenates (prepared in 4.2. Western blot assay) were pipetted together with a mixture of two solutions: 40 µL of 20% trichloroacetic acid solution with a 320 µL of TBARS reagent (37 mmol/L C4H4N2O2S, 500 mmol/L NaOH, 15% *v*/*v* CH3COOH). After the incubation and cooling step, n-butanol and pyridine (14:1, *v*/*v*) were added and centrifuged 10 min at 5000× *g*. The resulting organic phase was measured at 535 nm by a Synergy H1 Hybrid Multi-Mode Microplate Reader (Biotek, VT, USA).

Based on the calibration curve from tetrabutylammonium malondialdehyde salt, the concentration of malondialdehyde (MDA) was calculated.

### 4.7. Statistical Analysis

Differences between groups were evaluated using one-way ANOVA and Bonferroni multiple comparison tests. Kolmogorov–Smirnov normality test to examine if variables are normally distributed was used. Data were expressed as mean ± SD; *p* < 0.05 was considered to be statistically significant.

## Figures and Tables

**Figure 1 ijms-24-03490-f001:**
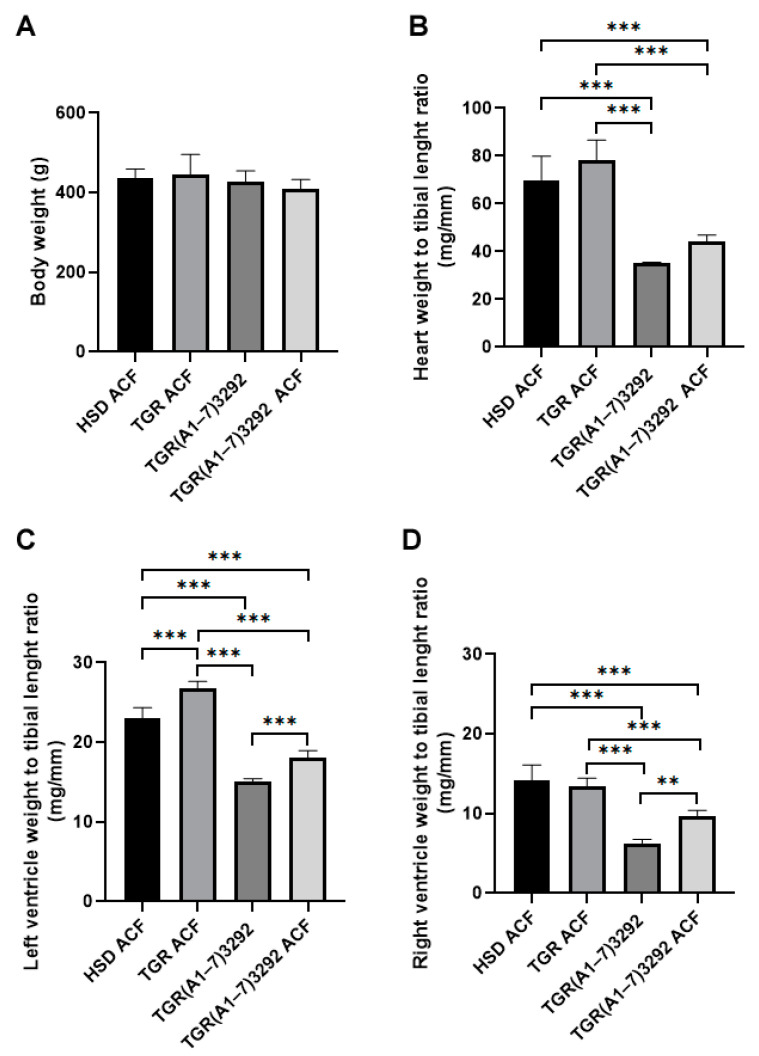
Biometric parameters of experimental rats. (**A**) Body weight, (**B**) heart weight to tibia ratio, (**C**) left ventricle weight to tibia ratio, (**D**) right weight to tibia ratio. Values are presented as mean ± SD (n = 10), ** *p* < 0.01, *** *p* < 0.001. ACF—aortocaval fistula; HSD ACF—Hannover Sprague–Dawley rats with ACF; TGR ACF—(mRen-2)27 transgenic rats with ACF; TGR(A1-7)3292—transgenic rats with increased expression of Ang (1-7); TGR(A1-7)3292 ACF—transgenic rats with increased expression of Ang (1-7) with ACF.

**Figure 2 ijms-24-03490-f002:**
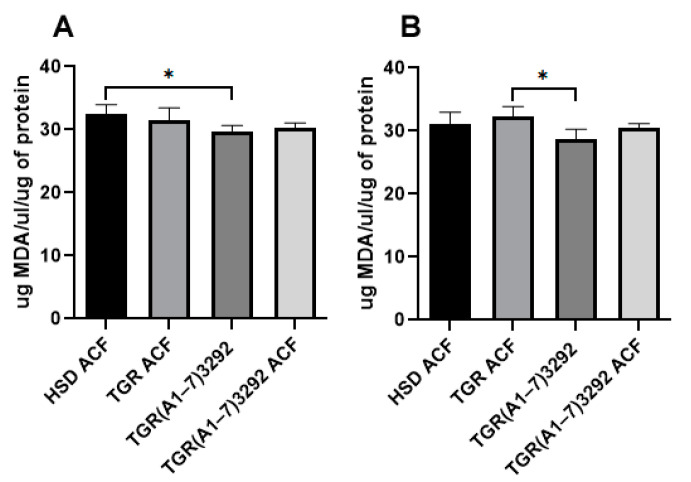
Reactive forms of thiobarbituric acid in the left (**A**) and right (**B**) heart ventricles. Values are normalized to the amount of protein per sample and presented as mean ± SD (n = 5), * *p* < 0.05. ACF—aortocaval fistula; HSD ACF—Hannover Sprague–Dawley rats with ACF; TGR ACF—(mRen-2)27 transgenic rats with ACF; TGR(A1-7)3292—transgenic rats with increased expression of Ang (1-7); TGR(A1-7)3292 ACF—transgenic rats with increased expression of Ang (1-7) with ACF.

**Figure 3 ijms-24-03490-f003:**
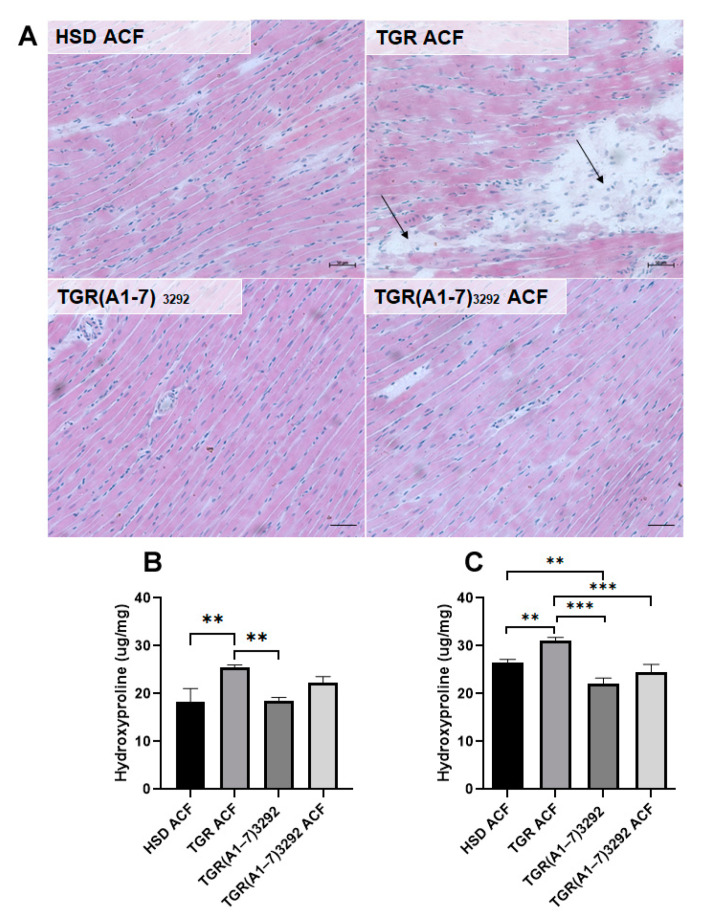
(**A**) Hematoxylin–eosin staining revealed the enlarged cardiomyocytes population and fibrosis (arrows) in TGR groups in heart apex. Scale bar represents 50 µm. Hydroxyproline content in the left (**B**) and right heart ventricles (**C**). Values are normalized to the weight of tissue used for analysis and presented as mean ± SD (n = 5), ** *p* < 0.01, *** *p* < 0.001. HSD ACF—Hannover Sprague–Dawley rats with ACF; TGR ACF—(mRen-2)27 transgenic rats with ACF; TGR(A1-7)3292—transgenic rats with increased expression of Ang (1-7); TGR(A1-7)3292 ACF—transgenic rats with increased expression of Ang (1-7) with ACF.

**Figure 4 ijms-24-03490-f004:**
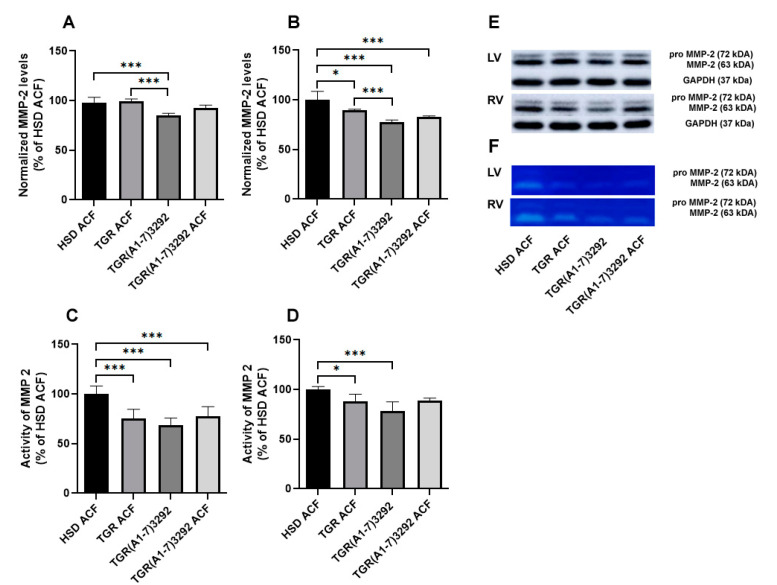
Quantitative evaluation of MMP-2 protein expression in the left (**A**) and right (**B**) ventricles and quantitative evaluation of MMP-2 activity in the left (**C**) and right ventricles (**D**). Representative Western blot (MMP-2) (**E**) and representative zymogram (MMP-2 activity) (**F**). Values are presented as mean ± SD (n = 5), * *p* < 0.05, *** *p* < 0.001. ACF—aortocaval fistula; MMP-2—matrix metalloproteinase 2; LV—left ventricle; RV—right ventricle; HSD ACF—Hannover Sprague–Dawley rats with ACF; TGR ACF—(mRen-2)27 transgenic rats with ACF; TGR(A1-7)3292—transgenic rats with increased expression of Ang (1-7); TGR(A1-7)3292 ACF—transgenic rats with increased expression of Ang (1-7) with ACF.

**Figure 5 ijms-24-03490-f005:**
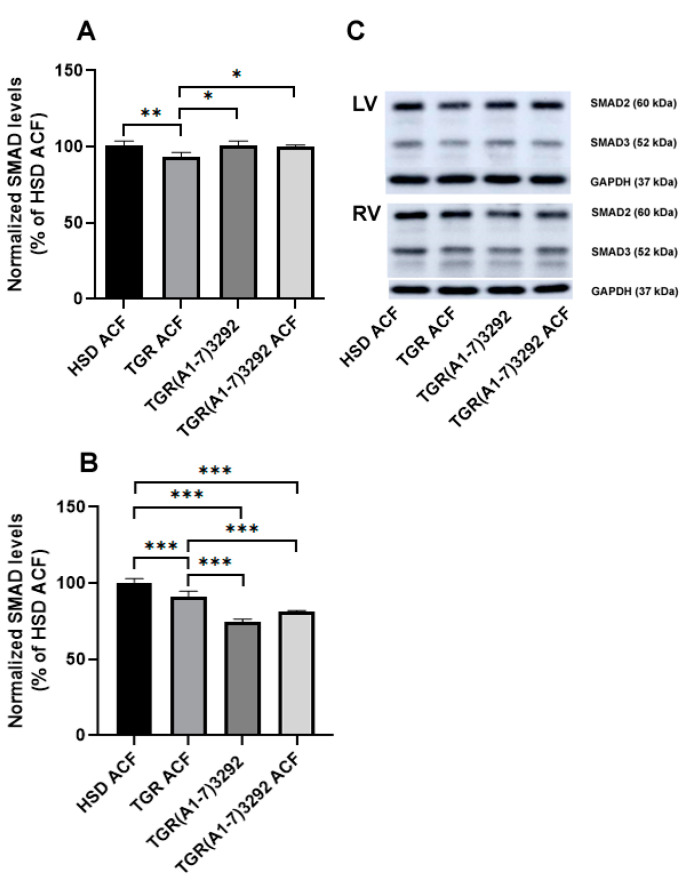
Quantitative evaluation of SMAD2/3 expression in the left (**A**) and right (**B**) ventricles and representative Western blot (**C**). Values of the sum of SMAD2 and SMAD3 are normalized to GAPDH and presented as mean ± SD (n = 5), * *p* < 0.05, ** *p* < 0.01, *** *p* < 0.001. ACF—aortocaval fistula; SMAD—intracellular extracellular signal transducing protein; GAPDH—glyceraldehyde-3-phosphate dehydrogenase; LV—left ventricle; RV—right ventricle; HSD ACF—Hannover Sprague–Dawley rats with ACF; TGR ACF—(mRen-2)27 transgenic rats with ACF; TGR(A1-7)3292—transgenic rats with increased expression of Ang (1-7); TGR(A1-7)3292 ACF—transgenic rats with increased expression of Ang (1-7) with ACF.

**Figure 6 ijms-24-03490-f006:**
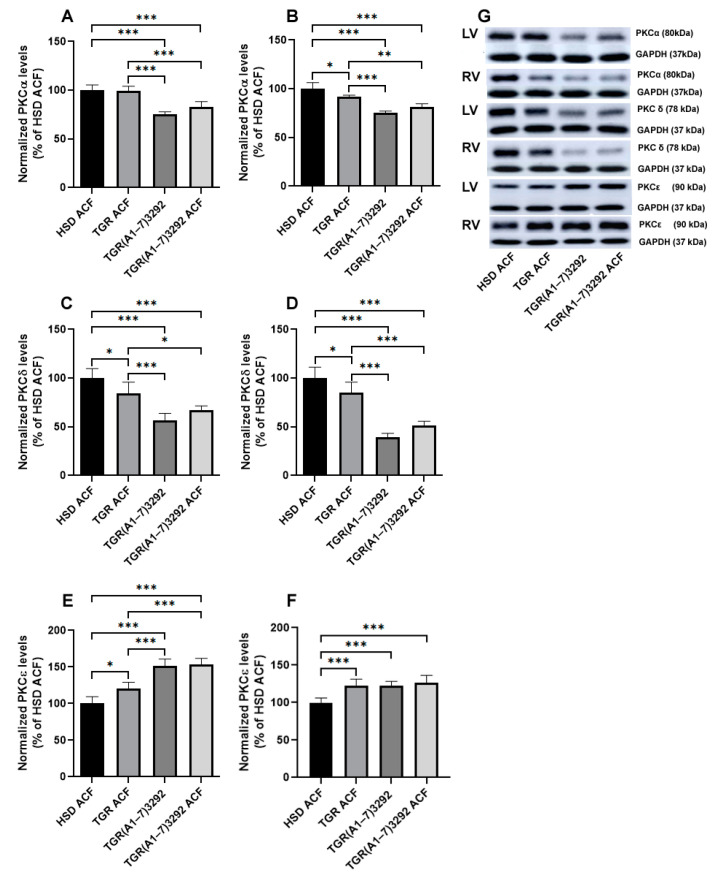
Quantitative evaluation of PKCα protein levels in the left (**A**) and right (**B**) ventricles, quantitative evaluation of PKCδ expression in the left (**C**) and right (**D**) ventricles, quantitative evaluation of PKCε expression in the left (**E**) and right (**F**) ventricles. Representative Western blot (**G**). Values are normalized to GAPDH and presented as mean ± SD (n = 5), * *p* < 0.05, ** *p* < 0.01, *** *p* < 0.001. ACF—aortocaval fistula; PKCα—protein kinase C alpha; PKCδ—protein kinase C delta; PKCε—protein kinase C epsilon; GAPDH—glyceraldehyde-3-phosphate dehydrogenase; LV—left ventricle; RV—right ventricle; HSD ACF—Hannover Sprague–Dawley rats with ACF; TGR ACF—(mRen-2)27 transgenic rats with ACF; TGR(A1-7)3292—transgenic rats with increased expression of Ang (1-7); TGR(A1-7)3292 ACF—transgenic rats with increased expression of Ang (1-7) with ACF.

**Figure 7 ijms-24-03490-f007:**
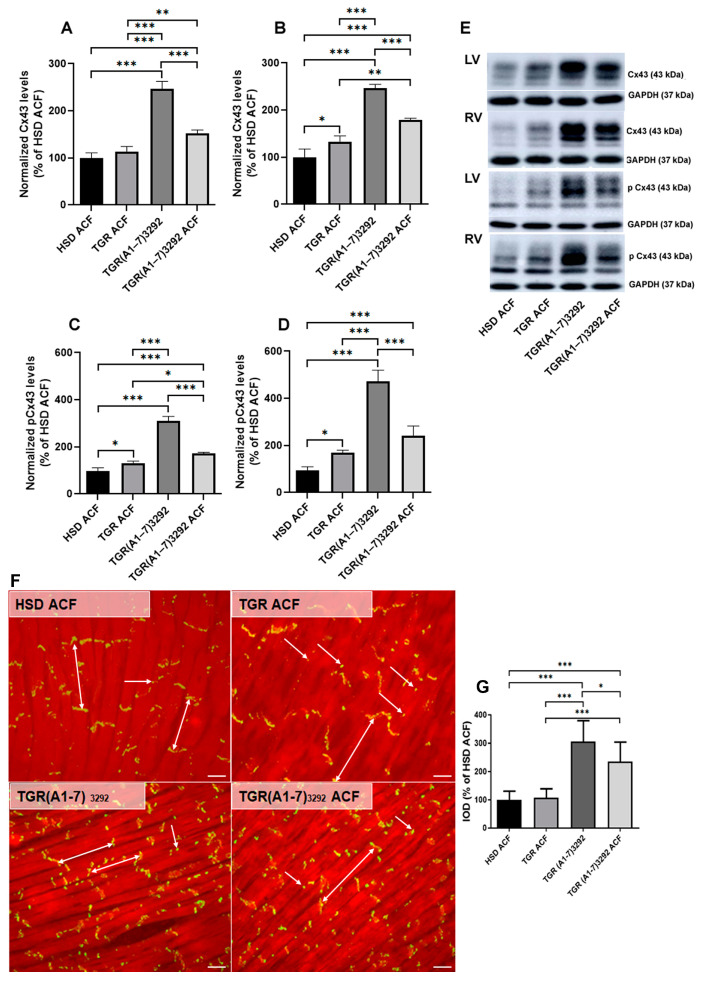
Quantitative evaluation of Cx43 expression in the left (**A**) and right (**B**) ventricles and quantitative evaluation of pCx43 expression in the left (**C**) and right (**D**) ventricles. Representative Western blot (**E**). Visualization of the myocardial connexin-43 (Cx43, green) and F-actin binding phalloidin (red) using immunofluorescence staining (**F**). Graphs represent a total integral optical density per area (IOD) of Cx43 (**G**). Values are normalized to GAPDH and presented as mean ± SD (n = 5), * *p* < 0.05, ** *p* < 0.01, *** *p* < 0.001, ACF—aortocaval fistula; Cx43—connexin 43; pCx43—phosphorylated connexin 43; GAPDH—glyceraldehyde-3-phosphate dehydrogenase; LV—left ventricle; RV—right ventricle; HSD ACF—Hannover Sprague–Dawley rats with ACF; TGR ACF—(mRen-2)27 transgenic rats with ACF; TGR(A1-7)3292—transgenic rats with increased expression of Ang (1-7); TGR(A1-7)3292 ACF—transgenic rats with increased expression of Ang (1-7) with ACF. Scale bar represents 200 µm. Double arrows represent the end-to-end localization of Cx43 and simple arrows show Cx43 localization on the lateral sides of the cardiomyocytes.

**Table 1 ijms-24-03490-t001:** Laboratory animals used in experiment.

Experimental Group	Sham Surgery (n)	ACF Surgery (n)
TGR(A1-7)3292	10	10
HSD	-	10
TGR	-	10

HSD—Hannover Sprague–Dawley rats; TGR—(mRen-2)27 transgenic rats; TGR(A1-7)3292—transgenic rats with increased expression of Ang (1-7); ACF—aortocaval fistula-induced congestive heart failure, n—sample size.

**Table 2 ijms-24-03490-t002:** Antibodies used for Western blot analysis.

Antibody	Dilution	Host	Type	Supplier/# Catalogue
Cx43	1:5000	Rabbit	Polyclonal	Sigma-Aldrich, St. Louis, MO, USA, #C6219
phos-^ser368^-Cx43	1:1000	Rabbit	Polyclonal	Santa Cruz Biotechnology, Dallas, TX, USA, #sc-101660
PKC-epsilon	1:2000	Rabbit	Polyclonal	Santa Cruz Biotechnology, Dallas, TX, USA, #sc-214
PKC-delta	1:2000	Rabbit	Polyclonal	Santa Cruz Biotechnology, Dallas, TX, USA, # sc-213
SMAD2/3	1:1000	Rabbit	Polyclonal	Cell Signaling Technology, Danvers, MA, USA, #3102
MMP2	1:500	Rabbit	Polyclonal	Santa Cruz Biotechnology, Dallas, TX, USA, # sc-10736
PKC α	1:500	Rabbit	Polyclonal	Santa Cruz Biotechnology, Dallas, TX, USA, # sc-208
GAPDH	1:1000	Rabbit	Polyclonal	Santa Cruz Biotechnology, Dallas, TX, USA, #sc-25778
Rabbit	1:2000	-	-	Cell Signaling Technology, Danvers, MA, USA, #7074

## Data Availability

The original Western blotting images can be found in the Appendix A.

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
