# Peer review of "Anti-Fibrotic Potential of Angiotensin (1-7) in Hemodynamically Overloaded Rat Heart"

_ijms, 2023, doi:10.3390/ijms24043490_

Round 1
Reviewer 1 Report
The authors, Sýkora et al., have submitted an original research article entitled "Anti-fibrotic potential of angiotensin (1-7) in hemodynamically two overloaded rat heart". In the manuscript, the authors have delineated the effect of Angiotensin (1-7) expression in the context of heart failure caused by aortocaval fistula at the molecular level. The authors found that various parameters indicated changes to the extracellular matrix of the ventricles with differential effects sometimes between the left and right ventricles. Analysis of the relevant regulators and signaling pathways showed that Angiotension (1-7) expression protects the heart from extracellular matrix remodeling through Cv43 phosphorylation. The following comments need to be addressed by the authors to improve the manuscript:
Major comments:
The authors could revise Table 1 for a graph to make it more visually appealing to the reader.
While it is interesting that the Left ventricle weight/Tibia in TGR(A1-7)3292 ACF rats were significantly lower than HSD ACF, it would be helpful to compare the levels reported by TGR(A1-7)3292 ACF rats with the levels reported by HSD mice.
In section 2.2, it would be helpful if the authors describe the importance of checking oxidative stress levels and ECM collagen accumulation in ACF rats and cite studies that recommend or have used TBARS levels or Hydroxyproline content as a marker for the respective parameters.
In section 2.2, the authors could determine if there is a significant reduction in the levels of hydroxyproline in TGR(A1-7)3292 ACF rats vs. TGR ACF rats. If so, this should be indicated and mentioned in the results. This would make it apparent that the expression of A1-7 helps to reduce the effect of TGR and bring it back to WT (HSD ACF) levels.
The authors should consider moving Figure 1 as a supplementary figure as it does not add any new dimension to the results. Also, Figures 2 and 3 could be combined into a single figure.
The authors should consider describing Figure 3 in a more nuanced way that will help to explain the effect of A1-7 expression as compared to TGR ACF rats.
In the description for figure 4, the authors should discuss the exciting finding that while there is a significant reduction in MMP2 activity in TGR or TGR(A1-7)3292 ACF rats as compared to HSD ACF rats, there is no corresponding decrease in protein levels unlike what is observed in the right ventricle, where a decline in activity can be directly attributed to decrease in protein levels. This indicates that the mechanism of regulation of MMP2 activity differs between the ventricles.
Figures 4 and 5 could be combined into a single figure.
In all western blot results, the authors should maintain homogeneity across the manuscript concerning how the representative western blots are depicted. In some sub-figures, the authors have shown that GAPDH was run on a separate gel while it was run on a single gel for other samples. This discrepancy should be justified. Also, in the quantification for all western blot results, since the levels of all proteins are normalized to GAPDH levels, the y-axis title should be Normalized ________ levels (% of HSD ACF).
In figure 6, the authors should denote if the graph is for SMAD2 or SMAD3 or total SMAD (SMAD2 + SMAD3).
In figure 7B, the levels of PKC-alpha appear to be similar in TGR ACF and TGR(A1-7)3292 ACF in the representative image. In contrast, the authors have shown a significant difference in the quantification. Please revise the representative image to reflect the quantification results.
Figures 7, 8, and 9 could be combined into a single figure as they all depict expression levels of different PKCs across the same treatment groups.
In Figure 9, the authors have shown that while TGR(A1-7)3292 ACF have a significant difference from the levels of HSD ACF and/or TGR ACF, the authors have not denoted similar significance for TGR (A1-7)3292 even though the latter have similar expression levels.
The representative blot in Figure 10A appears to have Cx43 run on different gels. However, the same is not reflected in the image. Also, it would be helpful to run the protein of interest and GAPDH on the same gel for all treatment groups or different gels for all treatment groups.
Figures 10, 11, and 12 should be merged into a single figure as they all discuss the different parameters of Cx43.
Author Response
Rev1
We are thankful for the valuable critical comments of the reviewer. In the revised version, we did our best to modify the manuscript according to the recommendations. The changes in the text are colour-marked.
The authors, Sýkora et al., have submitted an original research article entitled "Anti-fibrotic potential of angiotensin (1-7) in hemodynamically two overloaded rat heart". In the manuscript, the authors have delineated the effect of Angiotensin (1-7) expression in the context of heart failure caused by aortocaval fistula at the molecular level. The authors found that various parameters indicated changes to the extracellular matrix of the ventricles with differential effects sometimes between the left and right ventricles. Analysis of the relevant regulators and signaling pathways showed that Angiotension (1-7) expression protects the heart from extracellular matrix remodeling through Cv43 phosphorylation. The following comments need to be addressed by the authors to improve the manuscript:
Major comments:
The authors could revise Table 1 for a graph to make it more visually appealing to the reader.
Thank you for your comment, we changed the table with biometric parameters to a graph.
While it is interesting that the Left ventricle weight/Tibia in TGR(A1-7)3292 ACF rats were significantly lower than HSD ACF, it would be helpful to compare the levels reported by TGR(A1-7)3292 ACF rats with the levels reported by HSD mice.
Thank you for your comment, but the limitation of our experiment was the absence of control HSD and TGR, as we have stated at the end of the discussion.
In section 2.2, it would be helpful if the authors describe the importance of checking oxidative stress levels and ECM collagen accumulation in ACF rats and cite studies that recommend or have used TBARS levels or Hydroxyproline content as a marker for the respective parameters.
Thank you for your comment, we agree and added to the lines 89-93: Oxidative stress plays an important role in heart failure, including ACF, and new findings suggest a link between heart failure and oxidative stress. The increased synthesis and secretion of ROS and leads to a disturbance of the oxidative balance, which in turn stimulates many signaling pathways that regulate various processes, including the pro-motion of cell proliferation and migration and the secretion of ECM [19].
In section 2.2, the authors could determine if there is a significant reduction in the levels of hydroxyproline in TGR(A1-7)3292 ACF rats vs. TGR ACF rats. If so, this should be indicated and mentioned in the results. This would make it apparent that the expression of A1-7 helps to reduce the effect of TGR and bring it back to WT (HSD ACF) levels.
Thank you for your comment, we agree and added to the lines 103-111: Hydroxyproline is a breakdown product of collagen occurring mainly in tissue fibrosis and overall cardiac remodeling expected in ACF-induced heart failure [20]. Hydroxyproline content, a marker of ECM collagen accumulation, was increased in both heart ventri-cles in TGR ACF vs HSD ACF. However, there was a significant decrease of hydroxyproline content in the right ventricle of TRG(A1-7)3292 ACF vs TGR ACF (Figure 2). The sum of the results of HE staining and biochemical analysis of hydroxyproline points to the fact that ANG1-7 in our HF model significantly reduces the level of fibrosis, especially in the right ventricle of the heart compared to TGR and normalizes them to the level of HSD control.
The authors should consider moving Figure 1 as a supplementary figure as it does not add any new dimension to the results. Also, Figures 2 and 3 could be combined into a single figure.
Thank you for your comment, we kept the figure with the TBARS result in the article due to the importance of measuring the oxidation state parameter in HF models: Lines 89-93: Oxidative stress plays an important role in heart failure, including ACF, and new findings suggest a link between heart failure and oxidative stress. The increased synthesis and secretion of ROS and leads to a disturbance of the oxidative balance, which in turn stimulates many signaling pathways that regulate various processes, including the pro-motion of cell proliferation and migration and the secretion of ECM [19]. We combined figures 2 and 3 into one figure.
The authors should consider describing Figure 3 in a more nuanced way that will help to explain the effect of A1-7 expression as compared to TGR ACF rats.
We apologize for a not clear written description of figure 3. As You recommended in the previous comment, we combined Figures 2 and 3 where it is more obvious to see the antifibrotic effect of A1-7. Legend in the Figure 3 was rewritten (Lines 115-120).
Note, hematoxylin–eosin staining (Figure 3A) revealed in TGR ACF rats an increased focal area infiltrated with polymorphonuclears (arrows), the histopathological feature of the hyper- or hypothyroid status. Besides, hydroxyproline evaluation of collagen deposition (Figure 3B, C) revealed increased levels of collagen content in TGR ACF rats. A1-7 notably suppressed polymorphonuclears and hydroxyproline content in TGR ACF rats indicating its antifibrotic potential.
In the description for figure 4, the authors should discuss the exciting finding that while there is a significant reduction in MMP2 activity in TGR or TGR(A1-7)3292 ACF rats as compared to HSD ACF rats, there is no corresponding decrease in protein levels unlike what is observed in the right ventricle, where a decline in activity can be directly attributed to decrease in protein levels. This indicates that the mechanism of regulation of MMP2 activity differs between the ventricles.
Thank you very much for Your interesting comment. We definitely agree that the mechanism of MMP-2 could differ between ventricles whether they have a different function. These changes could be also more pronounced in our case because the right ventricle is much more affected by hemodynamic overload. Unfortunately, the tendency of protein is slightly similar or with no change (HSD ACF vs TGR ACF), and is not going in the opposite direction, so it's hard to attribute it to a different mechanism. This supports also our previous experiments, where we did not find a different trend of MMP-2 between ventricles. We assume that this discrepancy between the ventricle could be attributed to the higher hemodynamic effect on right ventricle.
Figures 4 and 5 could be combined into a single figure.
Thank you for your comment, we combined figures 4 and 5 into one figure.
In all western blot results, the authors should maintain homogeneity across the manuscript concerning how the representative western blots are depicted. In some sub-figures, the authors have shown that GAPDH was run on a separate gel while it was run on a single gel for other samples. This discrepancy should be justified. Also, in the quantification for all western blot results, since the levels of all proteins are normalized to GAPDH levels, the y-axis title should be Normalized levels (% of HSD ACF).
Thank you for your comment, whenever possible, we measured the expression of the normalizing protein on the same membrane as the analyzed protein, unfortunately this is not possible with connexins, because they are located at approximately the same place 43 or 37 kDa, therefore we analyzed GAPDH for connexins on a second, equally pipetted membrane. We have renamed the y-axis of all western blot graphs according to your recommendations.
In figure 6, the authors should denote if the graph is for SMAD2 or SMAD3 or total SMAD (SMAD2 + SMAD3).
Thank you for your comment, we added to the description of figure 5 that it is the sum of SMAD2 and SMAD3.
In figure 7B, the levels of PKC-alpha appear to be similar in TGR ACF and TGR(A1-7)3292 ACF in the representative image. In contrast, the authors have shown a significant difference in the quantification. Please revise the representative image to reflect the quantification results.
Thank you for your comment, we replaced the representative blot for PKC-alpha with one more corresponding to the graphic representation.
Figures 7, 8, and 9 could be combined into a single figure as they all depict expression levels of different PKCs across the same treatment groups.
Thank you for your comment, we combined figures 7, 8 and 9 into one figure.
In Figure 9, the authors have shown that while TGR(A1-7)3292 ACF have a significant difference from the levels of HSD ACF and/or TGR ACF, the authors have not denoted similar significance for TGR (A1-7)3292 even though the latter have similar expression levels.
Thank you for your comment, we also added a comparison between HSD/TGR ACF and TGR(A1-7)3292 group to all graphs.
The representative blot in Figure 10A appears to have Cx43 run on different gels. However, the same is not reflected in the image. Also, it would be helpful to run the protein of interest and GAPDH on the same gel for all treatment groups or different gels for all treatment groups.
Thank you for your comment, we replaced a representative blot for Cx43 in the left ventricle with one that more closely reflects the statistical evaluation.
We analysed the normalizing protein GAPDH on the same gel/membrane, except for the analysis of Cx43, where we had to analyze GAPDH on another, similarly pipetted gel due to the similar molecular weight of ~43kDa or ~37kDa.
Figures 10, 11, and 12 should be merged into a single figure as they all discuss the different parameters of Cx43.
Thank you for your comment, we combined figures 10, 11 and 12 into one figure.

Reviewer 2 Report
The Authors prepared manuscript to show how thye explored ECM and connexin-43 (Cx43) signalling pathways in hemodynamically overloaded rat heart as well as the possible implication of 16 angiotensin (1-7) (Ang (1-7)) to prevent/attenuate adverse myocardial remodelling.
The work is interesting but I would like to raise some ways of improvement.
1. Please show in introduction more info about anti-fibrotic potential of angiotensin (1-7) in hemodynamically overloaded rat heart
2. Please explain a bit more differences between different rat strains (HSD ACF and TGR ACF strains). How different strain may affect the final data?
3. Using letters a and b in graphs is difficult to understand the meaning. I would like to change it into stars. Please flag p-value with star
Author Response
Rev2
We are thankful for the valuable critical comments of the reviewer. In the revised version, we did our best to modify the manuscript according to the recommendations. The changes in the text are colour-marked.
The Authors prepared manuscript to show how thye explored ECM and connexin-43 (Cx43) signalling pathways in hemodynamically overloaded rat heart as well as the possible implication of 16 angiotensin (1-7) (Ang (1-7)) to prevent/attenuate adverse myocardial remodelling.
The work is interesting but I would like to raise some ways of improvement.
- Please show in introduction more info about anti-fibrotic potential of angiotensin (1-7) in hemodynamically overloaded rat heart
Thank you for your comment, we agree and added to the lines 62-67: Studies have shown that the administration of Ang1-7, during hemodynamic overload of the heart reduce myocyte hypertrophy and cardiac fibrosis [14,15]. Deletion for ACE2, on the other hand, caused an increase in perivascular and interstitial fibrosis, ventricular di-latation, a decrease in intrinsic myocardial contractility, increased cardiac remodeling, and there was a general deterioration of ventricular functions up to increased mortality [16–18].
- Please explain a bit more differences between different rat strains (HSD ACF and TGR ACF strains). How different strain may affect the final data?
Thank you for your comment, we agree and added to the lines 391-401: The hypertensive (mRen-2)27 transgenic rat strain (TGR) shows strong expression of the murine Ren-2 transgene in extrarenal tissues, which induces and maintains hypertension through conventional angiotensin II (Ang II), and hypertension is readily controlled by in-hibition of the renin-angiotensin system (RAS ). This transgenic rat strain is now widely used to study a variety of conditions related to tissue RAS activation, including angiogen-esis, cytokine activation, profibrotic and inflammatory pathologies, thus contributing to the understanding of the underlying processes causing severe hypertension [50,51]. In the experimental model, 20 male TGR(A1–7)3292 laboratory rats, 10 male Hannover Spra-gue-Dawley (HSD) laboratory normotensive rats and 10 male (mRen-2)27 transgenic la-boratory hypertensive rats (TGR) at the age of 8 weeks were used. HSD rats are standardly used as a normotensive control for TGR rats.
- Using letters a and b in graphs is difficult to understand the meaning. I would like to change it into stars. Please flag p-value with star
Thank you for your comment, we corrected the graphs, we connected the significant differences between the groups with a bracket and marked the significance with a star. * P<0.05, ** P<0.01, *** P<0.001

Round 2
Reviewer 1 Report
The authors Sykora et al., have submitted the revised manuscript entitled "Anti-fibrotic potential of angiotensin (1-7) in hemodynamically overloaded rat heart." The authors have addressed most of the comments in the previous round of review satisfactorily. However, numerous other concerns remain before the manuscript can be accepted. Following are some of the concerns that the authors need to address to improve the scientific robustness of the manuscript:
In Figure 1, mentioning the parameter as part of the axis title of the Y-axis for each sub-figure will enhance the readability of the figure.
The statement on lines 89-90 is repetitive and could be revised to include the details only once.
The description for Figure 2 mentions no difference between the groups. The statement could be more nuanced as the levels reported by TGR(A1-7)3292 rats are significantly different but not when these rats undergo ACF.
In lines 103-105, the authors mention hydroxyproline as a breakdown product of collagen. However, the cited reference and description of this parameter in other parts of this manuscript mention hydroxyproline and collagen accumulation as markers of fibrosis. The authors should revise this contrasting text.
In line 108, the rat strain is incorrectly mentioned as TRG(A1-7)3292 instead of TGR(A1-7)3292. The authors should correct such inconsistencies throughout the manuscript.
The legend for figure 3A should include details on the area where cardiomyocytes are being depicted e.g. right or left ventricle. Accordingly, the details of the experiment need to be included in the materials and methods section.
The note in Figure 3 could be moved to the description of the results. Also, the accumulation of polymorphonuclears could be a consequence of fibrosis. Therefore, the effect of A1-7 on polymorphonuclears could be secondary while the effect on fibrosis and hydroxyproline content could be the primary one. The authors should discuss this possibility in the discussion section.
Like in section 2.2, including a short introduction about the parameter being tested and summarizing the results of all sub-figures helps the readers to better understand the significance of the study and appreciate the authors' efforts and knowledge of the topic. It would be helpful to revise all the result sections accordingly.
In section 2.3, the authors could include their response to my previous question as part of the results to justify the difference in protein levels in the left ventricle. It would be helpful to also note the sub-figures that demonstrate that the right ventricle had higher hemodynamic overload as compared to the left.
There is no labeling in all the original western blotting figures. Therefore, the following is my interpretation of the images: Across all western blotting figures, the two images are indicative of the left ventricle (depicted above) and right ventricle (depicted below), each with 14 samples per gel, which given 4 treatments groups, does not add up to n=10 rats/treatment group. The authors should address this discrepancy. Also, if single gels/membranes are cropped to separately probe for different antibodies, their representative images may not be combined.
In figure 4, it would be helpful to have the arrangement of the sub-figures and their description in the legend in alphabetic order, as with all other figures.
In Figure 4E, the images of MMP2 expression appear to be different as compared to the original blots of MMP2 included as part of Original Images for Blots/Gels. It would be helpful if the authors marked the lanes that were selected from the original blot for the representative image.
In section 2.4, the authors mention that there is no change in the protein levels of SMAD2/3 together in the left ventricle. However, Figure 5A shows significant differences in them between different treatment groups. The authors should revise the description based on the results.
In section 2.5, the authors need to include a reference article demonstrating the use of respective PKCs for the parameters mentioned.
On line 171, the statement should include reference to both PKCα and PKCδ as both of them show a reduction in both ventricles upon A1-7 over-expression.
The features described by the authors in lines 204 - 206 need to point out in the representative images to better appreciate these findings.
For Figures 7C and 7D, the phosphorylation of Cx43 has to be normalized to the total expression of Cx43 (band intensities of Cx43 + pCx43). The authors should revise these calculations.
Since increased PKCe in TGR(A1-7)3292 rats as compared to controls (Figure 6) did not result in increased Cx43 phosphorylation (Figure 7), the authors need to discuss this abnormally in the discussion.
Author Response
The authors Sykora et al., have submitted the revised manuscript entitled "Anti-fibrotic potential of angiotensin (1-7) in hemodynamically overloaded rat heart." The authors have addressed most of the comments in the previous round of review satisfactorily. However, numerous other concerns remain before the manuscript can be accepted. Following are some of the concerns that the authors need to address to improve the scientific robustness of the manuscript:
In Figure 1, mentioning the parameter as part of the axis title of the Y-axis for each sub-figure will enhance the readability of the figure.
Thank you for your comment, we agree, they renamed the Y axis in the biometric data charts.
The statement on lines 89-90 is repetitive and could be revised to include the details only once.
Thank you for your comment, we changed the sentence " Oxidative stress plays an important role in heart failure, including ACF, and new findings suggest a link between heart failure and oxidative stress." to the sentence "Oxidative stress plays an important role in heart failure, including ACF."
The description for Figure 2 mentions no difference between the groups. The statement could be more nuanced as the levels reported by TGR(A1-7)3292 rats are significantly different but not when these rats undergo ACF.
Thank you for your comment, we will expand the results section in chapter 2.2. o statistically significant results in both left and right ventricles.
In lines 103-105, the authors mention hydroxyproline as a breakdown product of collagen. However, the cited reference and description of this parameter in other parts of this manuscript mention hydroxyproline and collagen accumulation as markers of fibrosis. The authors should revise this contrasting text.
Thank you for your comment, we refined our statement about hydroxyproline. „Hydroxyproline is a breakdown product of collagen occurring mainly in tissue fibrosis and overall cardiac remodeling expected in ACF-induced heart failure [20].“
In line 108, the rat strain is incorrectly mentioned as TRG(A1-7)3292 instead of TGR(A1-7)3292. The authors should correct such inconsistencies throughout the manuscript.
Thank you for your comment, we apologize for our inattention, we tried to avoid it.
The legend for figure 3A should include details on the area where cardiomyocytes are being depicted e.g. right or left ventricle. Accordingly, the details of the experiment need to be included in the materials and methods section.
Thank you for your comment, in the description under figure 3, we added that the given analysis was performed on the apex and we also added it to chapter 4.4.
The note in Figure 3 could be moved to the description of the results. Also, the accumulation of polymorphonuclears could be a consequence of fibrosis. Therefore, the effect of A1-7 on polymorphonuclears could be secondary while the effect on fibrosis and hydroxyproline content could be the primary one. The authors should discuss this possibility in the discussion section.
Thank You very much, as You recommend, we added description in the results part. In this study we were not interested into polymorphonuclears examination. We just performed basic histological haematoxylin eosin staining for myocardial structure demonstration where we also detected obvious polymorphonuclears in TGR ACF group. For potential readers can be helpful and interesting to know that polymorphonuclears could be a consequence of fibrosis. Finding that A1-7 could have effect on polymorphonuclears is very interesting and it will be our aim in the future study to evaluate this invasion of cells, particularly polymorphonuclear neutrophils by different histochemical analysis. Whether we did not measure this parameter in this study, it is very hard to discuss and compare it with our measured parameters. This is very interesting finding, and we would like to support this statement with relevant results.
Like in section 2.2, including a short introduction about the parameter being tested and summarizing the results of all sub-figures helps the readers to better understand the significance of the study and appreciate the authors' efforts and knowledge of the topic. It would be helpful to revise all the result sections accordingly.
Thank you for your comment, we added a brief reason for our analysis, the given marker, to each result chapter.
In section 2.3, the authors could include their response to my previous question as part of the results to justify the difference in protein levels in the left ventricle. It would be helpful to also note the sub-figures that demonstrate that the right ventricle had higher hemodynamic overload as compared to the left.
We added our response also to result part 2.3.
The mechanism of MMP-2 could differ between ventricles whether they have a different function. These changes could be also more pronounced in our case because the right ventricle is much more affected by hemodynamic overload. Unfortunately, the tendency of protein is slightly similar or with no change (HSD ACF vs TGR ACF), so it is hard to attribute it to a different mechanism. We assume that this discrepancy between the ventricles could be attributed to the higher hemodynamic effect on right ventricle.
There is no labeling in all the original western blotting figures. Therefore, the following is my interpretation of the images: Across all western blotting figures, the two images are indicative of the left ventricle (depicted above) and right ventricle (depicted below), each with 14 samples per gel, which given 4 treatments groups, does not add up to n=10 rats/treatment group. The authors should address this discrepancy. Also, if single gels/membranes are cropped to separately probe for different antibodies, their representative images may not be combined.
Thank You very much for this comment based of which we found discrepancies in whole figure descriptions. As we stated in methods part, we had 10 rats per group in experiment. For methodological diversity, we have used for individual analysis tissue samples from 5 rats. This discrepancy is repaired and changed through the text.
We showed 2 original western blotting images for one protein in LV (28 samples per gel) and 2 original western blotting images for one protein in RV (28 samples per gel). Whether we calculated every protein to control (HSD ACF), there is higher number of samples.
In figure 4, it would be helpful to have the arrangement of the sub-figures and their description in the legend in alphabetic order, as with all other figures.
Thank you for your comment, we sorted all the descriptions under the figures alphabetically.
In Figure 4E, the images of MMP2 expression appear to be different as compared to the original blots of MMP2 included as part of Original Images for Blots/Gels. It would be helpful if the authors marked the lanes that were selected from the original blot for the representative image.
Thank you for your comment, all representative blots were made separately, and were not done as a pooling of band sections from the blots that we used for quantitative analysis.
In section 2.4, the authors mention that there is no change in the protein levels of SMAD2/3 together in the left ventricle. However, Figure 5A shows significant differences in them between different treatment groups. The authors should revise the description based on the results.
Thank you for your comment, we will expand the results section in chapter 2.4. o statistically significant results in left ventricles.
In section 2.5, the authors need to include a reference article demonstrating the use of respective PKCs for the parameters mentioned.
Thank you for your comment, in section 2.5. we have included references to selected PKCs pointing to their importance for HF pathology.
On line 171, the statement should include reference to both PKCα and PKCδ as both of them show a reduction in both ventricles upon A1-7 over-expression.
Thank you for your comment, we changed the sentence "Contrary to PKCδ, PKCε protein levels were increased in the TGR ACF and TRG(A1-7)3292 ACF rats vs HSD ACF as well as in TRG(A1-7)3292 ACF rats vs TRG ACF." and expanded it with a reference to PKCa to " Contrary to PKCα and PKCδ, PKCε protein levels were increased in the TGR ACF and TRG(A1-7)3292 ACF rats vs HSD ACF as well as in TRG(A1-7)3292 ACF rats vs TRG ACF."
The features described by the authors in lines 204 - 206 need to point out in the representative images to better appreciate these findings.
Thank you for your comment, we have added arrows to figure 7F, pointing both to the connexin located on the intercalated discs "end-to-end" and to the lateral connexin. The explanation can be found in the description under figure 7.
For Figures 7C and 7D, the phosphorylation of Cx43 has to be normalized to the total expression of Cx43 (band intensities of Cx43 + pCx43). The authors should revise these calculations.
Since increased PKCe in TGR(A1-7)3292 rats as compared to controls (Figure 6) did not result in increased Cx43 phosphorylation (Figure 7), the authors need to discuss this abnormally in the discussion.
We agree that some authors use ratio of phosphorylated connexin 43 to total connexin 43 to determine phosphorylated status. We also used to do it in our previous studies where we detected with one antibody connexin 43 with clearly divided three bands. According to molecular weight, first two bands were attributed to the phosphorylated forms, while the third lowest was considered to be the non-phosphorylated form (doi: 10.33549/physiolres.933413).
In this experiment and others (doi: 10.3390/biomedicines10071707, doi: 10.3390/antiox9060546) we were interested at connexin 43 phosphorylated at serine 368. We used special primary antibody to detect this form of Cx43. This phosphorylation leads to a reduction in gap junction permeability affecting intercellular communication. Several studies revealed regulation of Cx43 phosphorylation at S368 by multiple PKC isozymes, not only by PKC epsilon. For instance, PKCδ, a novel PKC isoform, has been shown to phosphorylate S368 leading to gap junction internalization and degradation through the proteasomal and lysosomal pathway (doi:10.1074/jbc.M113.533265; doi.org/10.3389/fcvm.2022.1080131). Weather in TGR(A1-7)3292 ACF rats (vsTGR ACFR) was PKC delta decreased, we can assume that increase Cx43 at serine 368 can be a result of decreased Cx43 internalization and degradation. We also added this information in the discussion part.

Round 3
Reviewer 1 Report
The authors Sykora et al., have submitted a revised manuscript entitled "Anti-fibrotic potential of angiotensin (1-7) in hemodynamically overloaded rat heart" as an original research article. The authors have done an excellent job of revising the manuscript and answering all the questions satisfactorily. A few minor comments could be addressed before accepting the manuscript:
On line 329, the authors mention that they did not detect alteration of TGF-β. However, none of the figures include any data on the levels of TGF-β. The authors need to have the data or revise the statement.
The revised manuscript has multiple errors in English that might need to be addressed to enhance the experience of reading the manuscript.
Author Response
We are grateful for the reviewer valuable critical comments. In the revised version, we tried to edit the manuscript according to the recommendations. Changes to the text are color-coded to track changes.
The authors Sykora et al., have submitted a revised manuscript entitled "Anti-fibrotic potential of angiotensin (1-7) in hemodynamically overloaded rat heart" as an original research article. The authors have done an excellent job of revising the manuscript and answering all the questions satisfactorily. A few minor comments could be addressed before accepting the manuscript:
On line 329, the authors mention that they did not detect alteration of TGF-β. However, none of the figures include any data on the levels of TGF-β. The authors need to have the data or revise the statement.
Thank you for your comment, apologies, we have removed TGF-β from the relevant sentences.
The revised manuscript has multiple errors in English that might need to be addressed to enhance the experience of reading the manuscript.
Thank you for your comment, we read the manuscript again and corrected the grammar and word errors we found.
